# Evaluation of the Severity of Malocclusion in Children with Osteogenesis Imperfecta

**DOI:** 10.3390/jcm11164862

**Published:** 2022-08-19

**Authors:** Manuel Joaquín De Nova-García, Fabiola Bernal-Barroso, Maria Rosa Mourelle-Martínez, Nuria Esther Gallardo-López, Montserrat Diéguez-Pérez, Gonzalo Feijoo-García, Laura Burgueño-Torres

**Affiliations:** Dental Clinical Specialties Department, Faculty of Dentistry, Complutense University of Madrid, 28040 Madrid, Spain

**Keywords:** Osteogenesis Imperfecta, malocclusion, discrepancy index, dental arch, development

## Abstract

Occlusion is the way in which the dental arches are related to each other and depends on craniofacial growth and development. It is affected in patients with Osteogenesis Imperfecta (OI) who present altered craniofacial development. The malocclusion present in 49 patients diagnosed with different types of OI aged between 4 and 18 was studied. The control group of healthy people was matched for age, sex, and molar class. To study the mixed and permanent dentition, the American Board of Orthodontics (ABO) discrepancy Index was applied. The primary dentition was evaluated with a Temporary Dentition Occlusion Analysis proposed for this study. The OI group obtained higher scores in the Discrepancy Index than the control group, indicating a high difficulty of treatment. The most significant differences were found in types III and IV of the disease. Regarding the variables studied, the greatest differences were found in the presence of lateral open bite in patients with OI, and in the variable “others” (agenesis and ectopic eruption). The analysis of primary dentition did not show significant differences between the OI and control groups. Patients with OI have more severe malocclusions than their healthy peers. Malocclusion is related to the severity of the disease and may progress with age.

## 1. Introduction

Osteogenesis Imperfecta (OI) is a group of hereditary diseases caused by a collagen defect, the most relevant clinical manifestations of which affect the bones (bone fragility and deformity) but are not limited to them. In more than 90% of patients, the disease is caused by a heterozygous mutation in either of the two genes that encode type I collagen (COL1A1 on chromosome 17 and COL1A2 on chromosome 7) [1,2,3]. Its clinical variability led to classification into four subtypes that reflects the severity of the disorder [4,5], although new subtypes of the disease are currently being added, involving more collagen-forming genes (generally recessive transmission) [6,7,8].

The presence of type I collagen in the dentin and its altered development explain the dental clinical manifestation of dentinogenesis imperfecta (type 1), the most studied oral manifestation. Other oral manifestations have also acquired relevance, such as malocclusion, the most frequent being Class III, which has variable prevalence but generally affects more than 60% of the OI population studied [9,10,11,12,13]. Its possible relationship with craniofacial growth, when this is also altered, may partly explain this variability as well as its higher prevalence in individuals who are most severely affected by the disease. This seems to be confirmed by the cephalometric findings of reduced (retrusive) maxillary sagittal development, prognathic mandible associated with anticlockwise rotation of anterior growth, and compensatory dentoalveolar changes, marked by a significantly lower Wits value and poor dentoalveolar development [3,14,15,16,17]. Similarly, an early genotype–phenotype correlation analysis points to a relationship between the severity of the malocclusion and the underlying mutation, and to some influence of age and gender, which is still to be verified [18,19].

Therapeutic management is challenging for the clinician due to the challenges posed by the malocclusion itself, as well as the potential adverse effects on the dental movement of the antiresorptive therapy received by the patients to be treated [12,20].

Recent studies evaluated the severity of malocclusion in OI patients using different epidemiological indices such as peer assessment rating (PAR), the discrepancy index (DI), the Dental Health Component-Index of Orthodontic Treatment Need (DHC-IOTN) and Dental Aesthetic Index (DAI). They concluded that the severity for the clinician significantly exceeds that observed in healthy control populations [21,22].

This study contributes to previous approaches by adding the study of the OI population in primary dentition, on which there are no published works. The aim is to gather data regarding the progression of severity. Furthermore, the control sample includes only class III malocclusion to avoid discrepant comparisons with “opposite” malocclusions (severe classes II).

## 2. Materials and Methods

This study was supported by the AHUCE Foundation (Spanish Foundation that supports OI research) based on a Collaboration Agreement with the UCM.

### 2.1. Study Sample

The study universe was initially made up of 113 patients with OI (from national associations for the disease—AHUCE and AMOI) who attended the Master in Pediatric Dentistry at the Faculty of Dentistry of the Complutense University of Madrid for dental check-ups. To be included in the study, all subjects had to meet a series of inclusion criteria which included diagnosis of OI; age 18 or under; complete medical and dental history; complete primary dentition for the sample of patients in primary dentition; complete photographic series carried out in a standardized way; and panoramic radiography from age 6. In addition, a subgroup of 25 individuals made up of 13 girls and 12 boys in mixed/permanent dentition followed a study protocol prior to orthodontic treatment, consisting of lateral teleradiography and cephalometric tracings. Informed consent was obtained from all patients.

The final OI sample consisted of 49 children aged between 4 and 18, with different types of OI and in different phases of dentition. Of them, 41 were in mixed and definitive dentition (20 girls and 21 boys, with a mean age of 11.61), and 8 in primary dentition. (Table 1).

### 2.2. Control Samples

For the study, one healthy control matched by age, gender and malocclusion was taken for each child with OI. Most of the patients with permanent/mixed dentition (25 participants) followed a study protocol prior to orthodontic treatment (lateral teleradiography and cephalometric tracings) in a Bucofacial Diagnosis Center. All patients had a photographic study of sufficient quality. In the selection of controls, the following inclusion criteria were taken into account: healthy boys and girls whose age, gender and malocclusion were similar to the OI sample.

Children whose quality of records (photographs) did not allow for correct visual diagnosis were excluded from the study.

The control sample in mixed/permanent dentition consisted of 35 patients with molar Class III (20 girls with mean age 11.28, and 15 boys with mean age 10.25) and 6 patients with molar Class I (1 girl aged 10, and 5 boys with a mean age of 12.25). In primary dentition, the control sample consisted of 5 girls (mean age 5.2) and 4 boys (mean age 4).

Both samples were similar to those of patients with OI.

### 2.3. Method

A descriptive, cross-sectional study was carried out to analyze the severity of malocclusion in a group of patients with OI in primary, mixed and permanent dentition in the Master in Pediatric Dentistry at the Faculty of Dentistry of the Complutense University of Madrid.

To evaluate the permanent and mixed dentition, the Discrepancy Index used by the American Board of Orthodontics (Figure 1) [23] was used to determine the difficulty of orthodontic treatment in a sample of clinical photographs and radiographs of children with OI as well as in a sample of healthy controls. Four degrees of difficulty were defined according to the score (low: 0–7 points; moderate-low: 8–15 points; moderate-high: 16–25, and high: >26 points).

To evaluate the primary dentition, an Occlusion Analysis in Temporary Dentition was designed by examining the dental arches and their occlusal characteristics in the transverse, vertical and sagittal planes (Figure 2). Different degrees of severity of malocclusion were also estimated according to the score (low: 0–7 points; medium: 8–14 points, and high:>15 points).

The radiographic and orthodontic studies turned out to be complementary tools for clinical examination, being useful for studying some alterations and for obtaining cephalometric data.

The clinical examination was performed by a single examiner. Intra-examiner efficacy was assessed by duplicate examinations, re-measuring 35% of the sample and using a Kappa statistical analysis, obtaining a Kappa concordance index of 0.97.

### 2.4. Statistical Method

For the statistical analysis, the results obtained from the sample in mixed and permanent dentition and the sample in primary dentition were studied separately.

A Kolmogorov–Smirnov Test was performed on the samples (NPAR TEST procedure) to determine if the quantitative variables of the study showed normal distribution. If not, the following were applied:Non-parametric Kolmogorov–Smirnov test with the Mann–Whitney-Wilcoxon test to compare the measurements of the quantitative variable between the control group and the OI Group.Non-parametric test for samples not related to the Kruskal–Wallis test to compare the quantitative variables between the control group and the different subgroups within the OI group (OI type I, OI type III, OI type IV and OI type V).

For comparison between the control group and the different types of OI (multiple means), Analysis of Variance (ANOVA) was performed. The study was completed with Student’s *t* test for the comparison of quantitative means between the control group and the OI group in order to obtain descriptive tables not present in non-parametric tests.

Statistical analysis of the data was carried out with the SPSS 25.0 software (SPSS, Inc., Chicago, IL, USA) for Windows.

## 3. Results

### 3.1. Mixed and Permanent Dentition

After applying the Discrepancy Index (ID) to the total sample and obtaining results for the different variables, in the OI group 68.3% of the sample (OI types III, IV and V) presented high scores (high degree of treatment difficulty), while the remaining 31.70% (OI type I) presented moderate-low treatment difficulty.

The total score of the Discrepancy Index is the sum of the variables included in the index. The final score indicates the severity of malocclusion. In the case of the OI group, the mean score was 30.59 points (high degree of treatment difficulty), while in the control group, the mean score was 13.27 points (moderate-low degree of treatment difficulty). Malocclusion is accentuated in the most severe forms of the disease (III and IV) (Figure 3), which showed the highest scores. The differences were statistically significant at 95% between the two groups (*p* = 0.001 in Student’s *t*-test and Mann–Whitney test *p* = 0.001).

After analysis of the different components, a great variability of the results was found depending on the type of OI (Table 2).

The occlusion analysis revealed Class I occlusion in 34.1%, Class II in 4.9% and Class III in 61% of the total OI sample. Depending on the type of OI, the prevalence of malocclusion also varied, with Angle class I prevalent in patients with OI type I (61.5%) and Angle class III in patients with OI types III and IV (77.8% and 75%, respectively).

The most significant differences between the OI and control groups were found in lateral open bite, which was present in OI types III and IV but absent in OI type I and the control group.

Inverted overjet was also more prevalent and accentuated in the OI sample, which contributes to the (significant) differences with the control group.

Posterior crossbite is more common in patients with OI where the maxilla is poorly developed transversely and the mandible shows altered growth in all spatial planes. In our sample, 46.3% presented bilateral posterior crossbite, 26.8% presented unilateral posterior crossbite, and 26.8% did not present any type of transverse alteration. Such alterations were most frequent in OI types III and IV (77.8% and 100%, respectively).

In the “others” section, the most significant differences were due to the more frequent presence of dental agenesis and ectopic eruptions in the OI group, which were much less frequent in the control group. Statistical analysis revealed that there were significant differences at 95% (*p* = 0.004 in Student’s *t* test and Mann–Whitney test *p* = 0.001).

The ANOVA test (between groups and types) revealed that the most significant differences between the control groups and different types of OI were found in lateral open bite and occlusion between control groups and OI types III and IV, and “Others” between control groups and OI type IV.

### 3.2. Subgroup Sample Analysis

In order to measure the severity of the malocclusion in both the subgroup sample and the control group, the ABO Index (American Board of Orthodontics) was used. In this index, there is a section in which cephalometric angles are measured, and the severity of the malocclusion is elucidated by the results.

The use of these angles in orthodontics as a measure of the severity of malocclusion is because the ANB angle forms part of the classic cephalometric diagnosis (Steiner analysis) of malocclusions (skeletal analysis) and provides us with a first approach for analyzing the intermaxillary skeletal relationship (in the anteroposterior direction). It is one of the indicators for the differential diagnosis of the skeletal/dental class.

The other two angles add diagnostic accuracy to class III malocclusion. The IMPA angle provides guidance on the position of the mandibular incisor relative to its bony base and is useful for differential diagnosis in class III malocclusions. When the cause is mandibular prognathism (in osteogenesis imperfecta (OI)), the lower incisors may present a retroclination reducing the angle of the incisors to the mandibular plane, while in class III malocclusions with a dental cause (pseudoclass III), the lower incisors maintain a normal or proclined position. The Go-Gn-SN angle, derived from the Bjork–Jarabak analysis, links the base of the mandible with the base of the skull and indicates the direction of growth. Its relevance among craniofacial characteristics in OI was highlighted by Waltimo-Sirén J. et al. [16]. They considered that the angle should be increased in response to a depressed position of the sella turcica, unless counterclockwise mandibular growth occurs, which is very common in OI. It could therefore indicate this mandibular growth trend.

When analyzing the differences in cephalometric data between the subgroup sample (OI) and the control group, significant differences were observed for the SN-GoGn angle, pointing to differences in the direction of craniofacial growth (Table 3).

### 3.3. Primary Dentition

A total of eight patients (5 girls and 3 boys, with a mean age of 5.40) were in complete primary dentition. Distribution by type of OI was as follows: OI type I (*n* = 5; 31.25%); OI type III (*n* = 1, 6.25%); OI type IV (*n* = 2; 12.5%).

Half of the sample presented interincisive diastemas and 25% primate spaces. The other 50% of the sample had interincisive crowding.

The incisal relationship shows that 25% of the sample had an increased overjet, 12.5% an inverted overjet, and the same percentage had an increased overbite. The highest percentage was observed in the anterior open bite (37.5%). A total of 12.5% did not present alterations in the incisal relationship.

Regarding the sagittal plane, half of the sample presented a straight terminal plane or short mesial step (25% each). The remaining 50% presented a distal step. For the canine class, 50% had Class I, 37.5% were of Class II, and the remaining 12.5% were Class III.

In the transverse plane, 37.5% presented with unilateral posterior crossbite and 12.5% with bilateral posterior crossbite. In the rest of the sample, no alterations were found at this level (Figure 4).

A comparative analysis of primary dentition between the OI and control groups revealed no statistically significant differences in any of the values covered by the Occlusion Analysis in Temporary Dentition (Table 4).

## 4. Discussion

### 4.1. Permanent and Mixed Dentition

Few studies have evaluated the severity of malocclusion and those that exist used different indices in small samples of patients with different types of OI. In all the studies, the most serious types of OI (III and IV) are overrepresented and their percentages exceed the expected prevalence in the OI population. This might be why they find the most severe malocclusions, for which demand for dental care is particularly great. Although the present study followed the norm, it included the highest percentage of children with OI type I, although it also had the highest percentage of OI type III (Table 5).

Studies in the literature used different methodologies including model analysis (Rizkallah, Jabbour), clinical examination (Nguyen), and photographs (present study). Even so, the key results are similar. The discussion therefore focuses on these results [19,21,22].

The results obtained coincide with all previous studies in finding more severe malocclusions in the OI samples, indicated by the high total scores in all the indices applied. Our total score for the ID index (30.56) was slightly higher than that obtained by Rizkallah et al. (29.8) [21]. This means that 68.3% of our OI sample (types III, IV and V) presented a high degree of treatment difficulty (>26 points), while in the study by Rizkallah et al. 39% of the sample was above 31 points. The results obtained with other indices indicated similar conclusions. In the Rizkallah study, the PAR index was above 31 points for 53% of the OI patients. The study by Nguyen et al. in 2017, with a DHC-IOTN index of 89% of the OI sample, and with a DAI index of 61.5%, showed the need for orthodontic treatment [22].

On the other hand, the severity of malocclusion and the difficulty of treatment in the control group is in the moderate-low range, showing significant differences. This could be related to the different control samples and the types and degrees of severity of malocclusion in them. Our study revealed smaller differences with the control samples than those of Nguyen et al. In the latter study, the control groups indicated a smaller selection bias and lower prevalence of malocclusion [22]. There are also minor differences with regard to the study by Rizkallah et al. in which the samples were paired by molar class and malocclusion [21].

By analyzing the components of the different indices applied, coincidences and discrepant results appeared. The most relevant coincidences were found in the components considered to exacerbate malocclusion in patients with OI, including anterior and lateral openbites, posterior crossbites, anterior crossbite and occlusion. In all of them, both the Rizkallah study and the present work, with the same index (DI), the greatest differences were found with the control group. These differences were confirmed with other indices (PAR, DHC-IOTN) that assessed similar components [21].

It is important to highlight an exception related to the occlusion component. Although all studies confirmed that class III malocclusion is the most prevalent in the OI samples—61% (the current study), 57% (Rizkallah et al.), and 73.1% (Nguyen et al.), in the present study, the differences with the control group were not significant due to their pairing with the study sample in order to homogenize both samples with respect to the type of malocclusion. This made it possible to compare other components of the DI index, such as incisal relationships (especially overjet), more correctly [21,22]. In this study, the differences shown in overjet between the OI sample and the control groups are related to a higher prevalence of inverted overjet in the OI sample. Conversely, in the Rizkallah et al. study, a greater severity of this component (overjet) was found in the control sample and was attributed to the severity of the Class II malocclusions included [21].

In contrast to other authors such as Rizkallah, Nguyen, or Jabbour, no significant differences were found in the severity of posterior crossbites, which is due to the fact that the control sample presented a higher prevalence of Class III malocclusions, which also resulted in the greater presence of posterior crossbites.

The differences between the OI group and the control group in the “Others” component, which seem to also be confirmed in other studies, must be qualified. The present study, as in that by Rizkallah et al., relied on radiographic examination and attributed these differences to the higher prevalence of dental agenesis (Rizkallah) and ectopic eruptions (current study). In the study by Nguyen et al., results were based on clinical examination, and the differences were due to the absence of teeth of different etiology (extraction, delayed tooth eruption, hypodontia or abnormal odontogenesis), which may accentuate such differences.

The results presented in this study confirm that the severity of malocclusion increases in the most severe types of the disease, supporting its relationship with altered craniofacial development [13,14,15,16]. The cephalometric findings revealed significant differences in the SN-GoGn angle, indicating changes in the direction of the mandibular growth of the OI sample compared to controls.

### 4.2. Primary Dentition

In the case of the sample of OI patients in primary dentition, no comparisons could be established as no similar studies were found that analyze primary dentition independently.

For the study of malocclusion, an Analysis of Occlusion in Primary Dentition was applied, based on the aspects that have the greatest influence on malocclusion. This analysis evaluated the dental arches (crowding, diastema, primate spaces), problems in the transverse plane (posterior crossbite), in the sagittal plane (incisal, canine and molar relationship), and in the vertical plane (overbite, open bite anterior, lateral open bite).

The sample of OI patients showed few findings in primary dentition, which does not help to predict the characteristic malocclusion in the mixed-permanent dentition group (Class III). Thus, a sagittal analysis of the OI sample revealed a short mesial step in 25% of samples and canine class III in 12.5%, with an inverted incisal relationship in another 12.5% of the sample. In no case were significant differences found with the control group.

Transversally, posterior crossbite was present in 37.5% of samples unilaterally and 12.5% bilaterally. Although 40% of the sample presented malocclusion in the transversal plane, the differences with the control group did not reach levels of significance.

The sample of OI patients in primary dentition analyzed in this study was limited (eight patients) and had a different representation of OI types as follows: OI type I (*n* = 5; 62.5%), OI type III (*n* = 1; 12.5%), OI type IV (*n* = 2; 24.5%). This small sample size together with the greater representation of the mildest type of the disease might partly explain the lack of significant differences between the OI group and the control group. However, conversely, some of the findings (canine class III, inverted overjet and transverse malocclusion) might be exacerbated in the presence of altered craniofacial growth. These findings needs to be explored further.

## 5. Conclusions

The OI group in mixed and permanent dentition presented more severe malocclusions, thereby implying greater treatment difficulty compared to the control group.

The most severe malocclusions were included in OI types III and IV, with those of moderate-low treatment difficulty being included in type I.

The index variables that most influence the severity of malocclusion are lateral open bite, the SN-GoGn angle and “others” (agenesis, midline deviations and ectopic eruptions).

In primary dentition, Occlusion Analysis in Temporary Dentition did not reveal differences with the control group, with both groups presenting malocclusions with moderate treatment difficulty.

In addition, and based on our experience in the orthodontic treatment of children with OI, we considered that none of the indices used to study the severity of malocclusion in the OI population took into account the therapeutic difficulties that may be linked to the effects of the antiresorptive treatments on tooth movement. These effects were particularly evident when correcting the lateral open bites that were prevalent in this group.

## Figures and Tables

**Figure 1 jcm-11-04862-f001:**
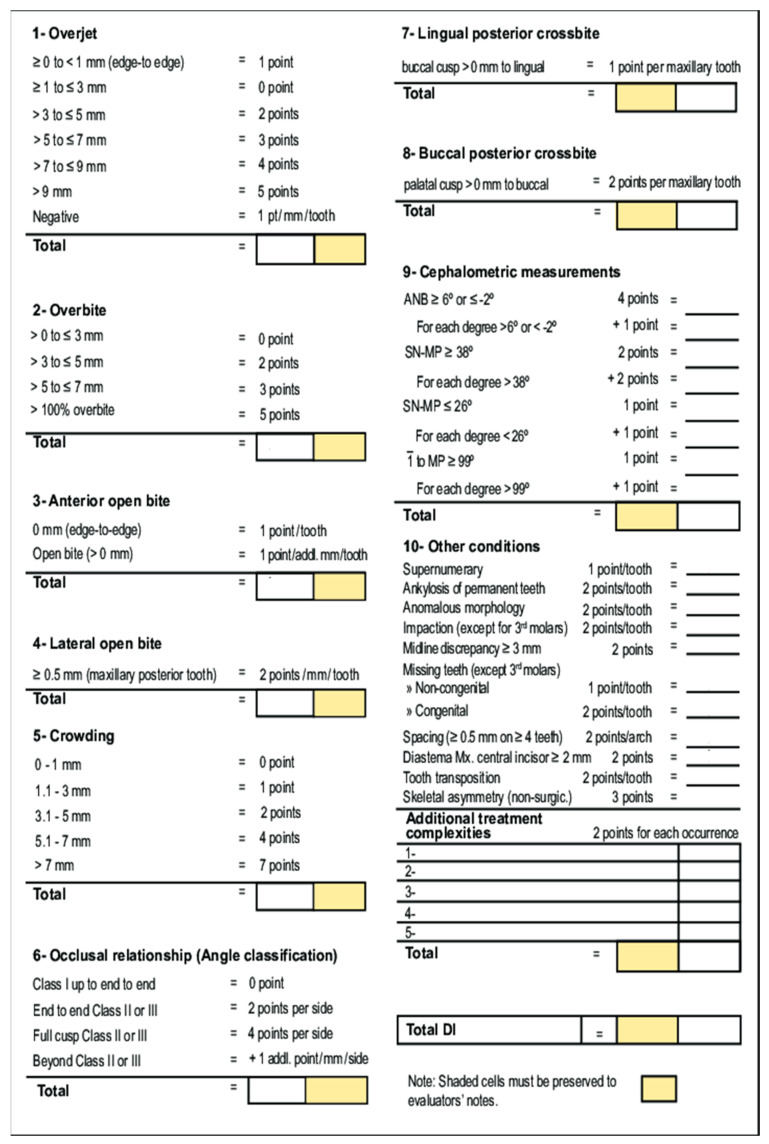
ABO Discrepancy Index [23]. American Board of Orthodontics (ABO); Analysis of the intermaxillary relationship (ANB); Sella Nasion and mandibular plane (SN-MP).

**Figure 2 jcm-11-04862-f002:**
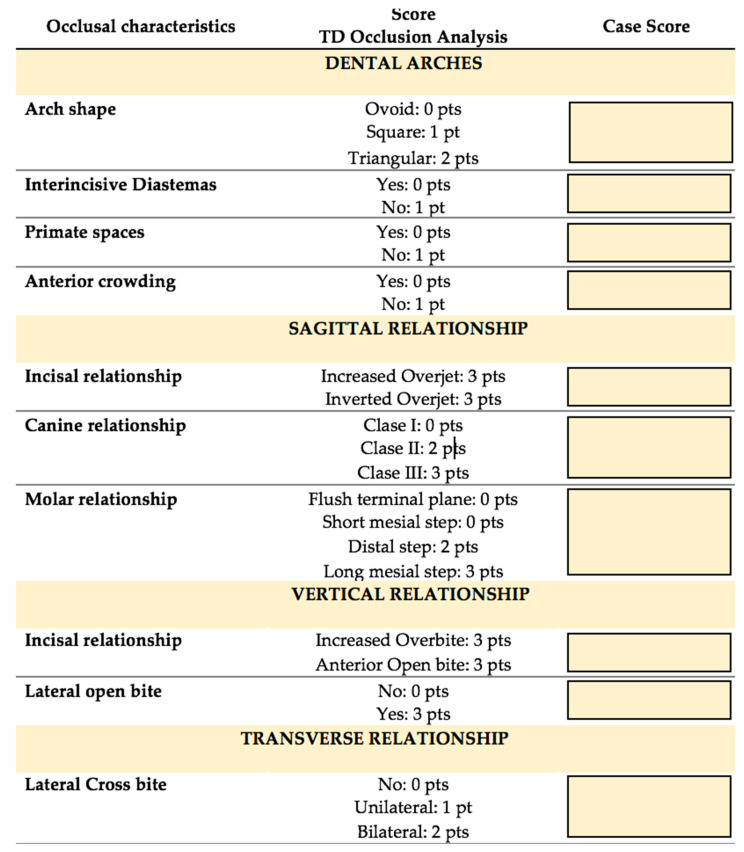
Primary Analysis Method.

**Figure 3 jcm-11-04862-f003:**
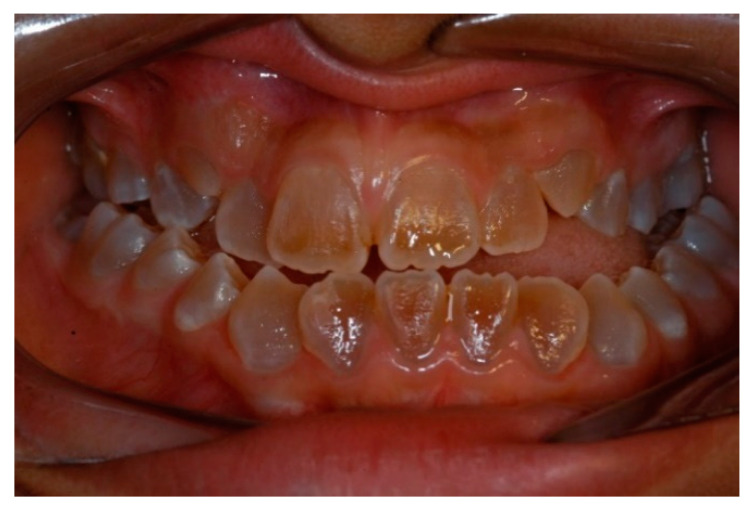
Malocclusion in OI type III patient and dentinogenesis imperfecta.

**Figure 4 jcm-11-04862-f004:**
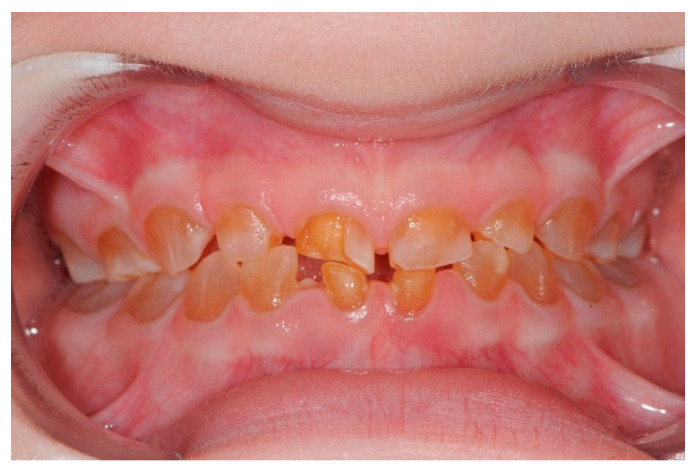
Temporal dentition in OI type I patient and dentinogenesis imperfecta.

**Table 1 jcm-11-04862-t001:** Distribution of the sample by severity of OI, phases of dentition and radiographic study.

Type of OI *	*n*	Female	Male	Mean Age	Dental Phases		Subgroup ^1^
					Mixed and Permanent	Primary	
I	18	10	8	8.67	13	5	10
III	19	11	8	12.11	18	1	9
IV	10	3	7	12.10	8	2	6
V	2	1	1	7.5	2	0	0
Total	49	25	24	11.6	41	8	25

^1^ Sample subgroup: lateral teleradiography and tracings. * Osteogenesis imperfecta.

**Table 2 jcm-11-04862-t002:** Comparative analysis of DI scores between OI group (total) and control group and OI types. Discrepancy index (DI); Osteogenesis Imperfecta (OI).

	OI (I)	OI (III)	OI (IV)	OI Total	Control
Total	14.69	35.5	45.2	30.56 ** (26.6)	13.27 ** (11.04)
Overjet	3.23	8.67	2.75	5.59 * (9.4)	2.32 * (3.1)
Overbite	0.4	0.6	0.8	0.61 (1.2)	0.34 (0.7)
Anterior open bite	2.23	3.72	2.31	3.56 (6.7)	1.66 (4.3)
Lateral open bite	2.46	11.22 **	27.75 **	11.12 * (20.2)	0.83 * (2.2)
Crowding	2.12	1.06	1.44	1.54 (3.2)	1.9 (2.1)
Occlusion	1.38	4.44 **	3.87 **	3.2 (3.2)	2.1 (2.1)
Lingual posterior crossbite	2.2	2.8	3.79	2.93 (3.2)	3.68 (6.5)
Bucal posterior crossbite	0.0	0.2	0.1	0.1 (0.4)	0.0 (0.0)
Other	1.38	1.39	4.25 **	1.93 * (2.9)	0.44 * (0.9)

Results are given as mean (SD). * Significant group differences, *p* < 0.05 with independent *t* tests. ** Significant group differences, *p* < 0.05 ANOVA test.

**Table 3 jcm-11-04862-t003:** Analysis of the cephalometric results in the subgroup sample. Osteogenesis Imperfecta (OI).

	OI	Control
ANB *** Angle	0.96 (1.7)	0.32 (1.10)
SN-GoGn **** Angle	3.24 * (3.7)	0.44 * (1.16)
IMPA **	0.68 (1.67)	0.40 (1.38)

Results are given as means. * Significant group differences, *p* < 0.05 with independent *t* tests. ** Incisal Mandibular Plane Angle. *** Analysis of the intermaxillary relationship. **** Sella-Nasion-Gonion-Gnation angle.

**Table 4 jcm-11-04862-t004:** Results of comparative analysis of occlusion in primary dentition in OI and control groups. Osteogenesis Imperfecta (OI).

	OI	Control
Total	8.25	9.38
Arch shape	0.63	0.50
Interincisive diastemas	0.50	0.13
Primate spaces	0.25	0.25
Incisive Crowding	0.50	0.50
Incisive Relationship	2.25	2.25
Canine Relationship	1.13	1.38
Molar Relationship	1.25	1.88
Lingual posterior crossbite	0.38	0.25
Bucal posterior crossbite	0.25	0.75
Anterior crossbite	1.13	1.50

Results are given as means.

**Table 5 jcm-11-04862-t005:** Characteristics of the study population: mean age, sex and type of Osteogenesis Imperfecta. Comparison with other studies. DI (Discrepance Index); PAR (Peer Assessment Rating); DAI (Dental Aesthetic Index); IOTN (Index of Orthodontic Treatment Need).

	Study Index	*n*	Male	Female	Mean Age	OI (I) *n* (%)	OI (III) *n* (%)	OI (IV) *n* (%)	OI (V) *n* (%)	OI (VI) *n* (%)
Present study	DI	41	21	20	11.61	13 (31.7)	18 (43.9)	8 (19.5)	2 (4.8)	
Rizkallah et al. (2013) [21]	DI and PAR	49	21	28	10.7	8 (16.3)	11 (22.4)	26 (53.1)	2 (4.0)	2 (4.0)
Nguyen et al. (2017) [22]	DAI and IOTN	26	18	8	5–19	7 (26.9)	10 (38.4)	9 (34.6)		
Jabbour et al. (2018) [19]	PAR	49	21	28	10.7	7 (14.2)	11 (22.4)	27 (55.1)	2 (4.0)	2 (4.0)

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
