# Peer review of "Evaluation of the Severity of Malocclusion in Children with Osteogenesis Imperfecta"

_jcm, 2022, doi:10.3390/jcm11164862_

Round 1
Reviewer 1 Report
The manuscript titled “Evaluation of the severity of malocclusion in children with Osteogenesis Imperfecta” assessed the severity of malocclusion of patients with osteogenesis imperfecta (OI) in primary, mixed and permanent dentition.
In this study, the authors determined that patients with OI had more severe malocclusions than their healthy peers. This is an interesting observation to prompt clinicians to focus on the relationship between occlusion and craniofacial growth and development. However, additional explanations are required to further illustrate the features of malocclusion of patients with OI.
Major concerns:
1. The authors claimed the control sample included only class III malocclusion, however, the actual control samples also included class I. The reason why class II was excluded might be explained further since previous studies also contained them.
2. In methods, different indexes were used to evaluate dentition. The message of raters should be added and if above two raters (including two) evaluated one same patient, intraclass correlation coefficients should be added.
3. In subgroup sample analysis, authors analyzed differences in cephalometric data between the subgroup sample and the control group using three angles (ANB Angle, SN-GoGn Angle and IMPA), the reason why these three angles were selected and the meaning of the results should be explained more specifically.
4. The mean age (1.61) of samples in the present study displayed in Table 5 was confusing.
Minor concerns:
1. Contents of tables and figures should be corrected.
2. The title of table 5 should be replaced correctly.
3. The format and punctuation should be double checked.
Author Response
Dear Reviewer:
The authors appreciate your comments and suggestions, these have helped improve the way we present our research. Below we will detail and resolve each of your comments.
Point 1:
The authors claimed the control sample included only class III malocclusion, however, the actual control samples also included class I. The reason why class II was excluded might be explained further since previous studies also contained them
Response 1: Regarding the control sample, our original idea was that the controls should manifest the same type of malocclusion (class III) as the individuals with OI (within the age and sex ranges studied). We believe that in this way the applied indices would better reflect not only the greater potential severity in the study sample, but also any more clearly involved factors when samples with a "homogeneous" malocclusion are compared. We therefore obtained a number of control individuals with class III malocclusion from an Orthodontic Diagnostic Center. Since class III malocclusion is not very prevalent in our country, we were not able to complete the control sample although, unlike other controls in other studies, in ours the proportion of individuals with class III malocclusion is much higher.
Point 2:
In methods, different indexes were used to evaluate dentition. The message of raters should be added and if above two raters (including two) evaluated one same patient, intraclass correlation coefficients should be added.
Response 2: The clinical examination was performed by a single examiner. Intra-examiner efficacy was assessed by duplicate examinations, re-measuring 35% of the sample and using a Kappa statistical analysis, obtaining a Kappa concordance index of 0.97.
This information has also been added in methods.
Point 3:
In subgroup sample analysis, authors analyzed differences in cephalometric data between the subgroup sample and the control group using three angles (ANB Angle, SN-GoGn Angle and IMPA), the reason why these three angles were selected and the meaning of the results should be explained more specifically.
Response 3: These aspects have been more fully explained in the results section.
In order to measure the severity of the malocclusion in both the subgroup sample and the control group, the ABO Index (American Board of Orthodontics) was used. In this in-dex, there is a section in which cephalometric angles are measured, depending on the re-sult, will be the severity of the malocclusion.
The use of these angles in orthodontics as a measure of the severity of malocclusion is because the ANB angle has formed part of the classic cephalometric diagnosis (Steiner analysis) of malocclusions (skeletal analysis) and provides us with a first approach for analysis of the intermaxillary skeletal relationship (in the anteroposterior direction). It is one of the indicators for the differential diagnosis of the skeletal/dental class.
The other two angles add diagnostic accuracy to class III malocclusion. The IMPA angle provides guidance on the position of the mandibular incisor relative to its bony base and is useful for differential diagnosis in class III malocclusions. When the cause is mandibular prognathism (in osteogenesis imperfecta (OI)), the lower incisors may present a retroclination reducing the angle of the incisors to the mandibular plane, while in class III malocclusions with a dental cause (pseudoclass III), the lower incisors maintain a normal or proclined position. The Go-Gn-SN angle, derived from the Bjork-Jarabak analysis, links the base of the mandible with the base of the skull and indicates the direction of growth. Its relevance among craniofacial characteristics in OI has been highlighted by Walti-mo-Sirén J. et al. (Am J Genet 2005). They consider that the angle should be increased in response to a depressed position of the sella turcica, unless counterclockwise mandibular growth oc-curs, which is very common in OI. It could therefore indicate this mandibular growth trend.
Point 4:
The mean age (1.61) of samples in the present study displayed in Table 5 was confusing.
Response 4: The average age in Table 5 is modified (11.61). This point is also included in the material and methods section. However, an error occurred in the table.
Point 5:
Contents of tables and figures should be corrected.
Response 5: In relation to figure 1, the meaning of the acronym ABO is indicated in the summary. The meaning of the acronym IMPA is shown in table 3. The meaning of the different acronyms used in table 5 have been previously explained in the introduction, where they are initially mentioned.
Point 6:
The title of table 5 should be replaced correctly.
Response 6: The following heading is added to Table 5: Characteristics of the study population: mean age, sex and type of Osteogenesis Imperfecta. Comparison with other studies.
Point 7:
The format and punctuation should be double checked.
Response 7: Thanks for your comment, some errors have been fixed.

Reviewer 2 Report
1. In the summary Authors use abbreviation which is not explain (ABO)
2. In explenation of the inclusion criteria Authors included the patients with "panoramic radiography from age 6". How the patients in age from 4 years can meet these criteria?
3. Table 5 – caption of the table is misleading
Author Response
Dear Reviewer:
The authors appreciate your comments and suggestions, these have helped improve the way we present our research. Below we will detail and resolve each of your comments.
Point 1:
In the summary Authors use abbreviation which is not explain (ABO).
Response 1: The meaning of the abbreviation ABO is added (Academy Board of Orthodontics).
Point 2:
In explanation of the inclusion criteria Authors included the patients with "panoramic radiography from age 6". How the patients in age from 4 years can meet these criteria?
Response 2:
Only panoramic radiographs were obtained in the mixed and permanent dentition sample.
Because Class III is not very prevalent in our country, all patients with this type of malocclusion in primary, mixed and permanent dentition in the Orthodontic Diagnostic Center database were included in the control sample, regardless of their age. For this reason, having a patient with Class III and temporary dentition was considered more important than her age.
Furthermore, in the case of the primary dentition sample, the records used in data collection did not include radiographic records.
This sample is part of a pilot study in which we observed occlusal characteristics in order to study the progression of occlusion, as well as growth and development in patients with Osteogenesis Imperfecta.
Point 3:
Table 5 – caption of the table is misleading
Response 3: The following heading is added to Table 5: Characteristics of the study population: mean age, sex and type of osteogenesis imperfecta. Comparison with other studies
